# Predicting $k$-Step Optimal Trajectories in Local Search

**Andoni I. Garmendia, Josu Ceberio, Alexander Mendiburu**
University of the Basque Country UPV/EHU, Spain
{andoni.irazusta,josu.ceberio,alexander.mendiburu}@ehu.eus

## Abstract

Local search is a key tool in combinatorial optimization. Once an operator is set, the method starts from a random solution and iteratively moves to a candidate neighbor. Typically, the best neighbor is sought, which requires visiting all neighbors, an approach that can be computationally expensive since it requires to recalculate the fitness function. Recently, neural networks have been employed with considerable success to predict the best move in a single shot, thereby reducing computational cost. However, this short-sighted approach, like traditional local search, tends to get stuck in local optima. To address this limitation, we propose neural models capable of predicting the optimal move after k local search steps, effectively learning the k-step trajectory that maximizes improvement in the objective function. Preliminary experiments on the Maximum Cut problem, which motivated this proposal, show that incorporating an imitation learning loss into the conventional reinforcement learning pipeline not only accelerates convergence but also achieves impressive performance, with 99% accuracy in selecting the optimal move within 3-step neighborhoods.

## 1   Motivation

Local Search (LS) methods Hoos and St$i\nu$tzle [2018] are a cornerstone in combinatorial optimization, where simple, iterative modifications, such as flipping a bit in a binary vector, are used to explore the solution space. Popular techniques like Tabu Search Glover [1989], and other metaheuristics Blum and Roli [2003] rely on LS as a fundamental building block to improve candidate solutions. However, these methods typically require evaluating a vast number of potential moves, which can be computationally demanding.

To overcome this limitation, recent advances have led to the development of neural network-based approaches that mimic the iterative improvement process Chen and Tian [2019], Wu et al. [2021], Garmendia et al. [2023]. We refer to these as Neural Improvement (NI) methods. Like classical LS methods, NI iteratively refines candidate solutions; however, NI methods directly choose the optimal move without exhaustively evaluating all neighboring candidates. The underlying neural network is trained on a set of instances and candidate solutions to propose modifications that enhance solution quality. During inference, these models can be applied to unseen instances to propose iterative modifications without the need for additional solution evaluation.

Predicting an optimal trajectory over $k$ steps is inherently more complex than predicting a single move. One-step NI methods often rely solely on reinforcement learning (RL) Sutton [2018], selecting actions based on reward signals corresponding to improvements in the objective function. Unfortunately, RL can be data-inefficient and slow to converge, a problem that is exacerbated when extending predictions to $k$-step trajectories, where the number of possible move sequences grows with respect to $k$.

XVI XVI Congreso Español de Metaheurísticas, Algoritmos Evolutivos y Bioinspirados (maeb 2025).

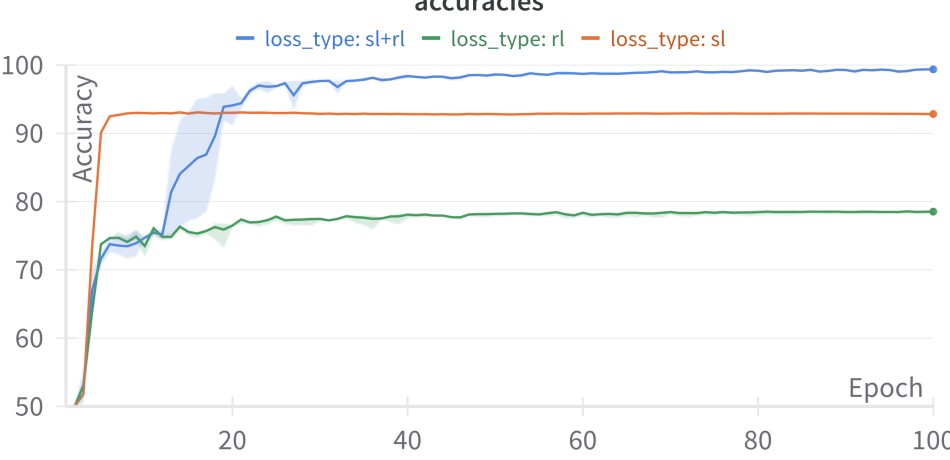

Figure 1: Accuracy of models where the Neural Improvement model is tasked with classifying the moves that lead to the greatest improvement in 3 steps on a Maximum Cut instances with Erdős–Rényi graphs of 20 nodes. The hybrid training (IL+RL) is performed with summing the two losses with equal weight.

To address these challenge, we propose a hybrid approach that integrates RL with imitation learning (IL) Hussein et al. [2017]. By computing optimal trajectories through exhaustive search on small instances during training, we generate explicit imitation labels that guide the learning process. This additional supervision has been shown to significantly accelerate convergence in other deep learning tasks Hester et al. [2018]. Our preliminary experiments on the Maximum Cut problem Goemans and Williamson [1995] support this idea. In fact, we have seen that combining RL and IL not only speeds up convergence but also substantially improves accuracy compared to using either method alone. As illustrated in Figure 1, our NI network predicts the move that best improves the objective within a 3-step neighborhood with 99% accuracy.

These impressive results motivate a deeper investigation into the mechanisms behind the performance gains achieved by combining RL and IL in the context of $k$-step trajectory prediction.

## 2 Hypothesis

Both RL and IL offer distinct advantages while also exhibiting inherent limitations when employed independently in training neural models for combinatorial optimization. *IL* accelerates training by providing explicit guidance through expert or optimal action labels; however, it may neglect near-optimal moves that, although not strictly optimal, are crucial for robust generalization across diverse problem instances. In contrast, *RL* rewards actions that are sufficiently good (even if not optimal), enabling the model to explore a broader range of promising moves. Nevertheless, RL often suffers from data inefficiency, as it requires extensive exploration in vast action spaces and tends to exhibit high variance during the early stages of training. This variance arises because many suboptimal actions are sampled before consistently promising trajectories are identified, frequently leading to premature convergence on suboptimal policies.

Based on these observations, we hypothesize that a hybrid training framework integrating both RL and IL will combine their complementary strengths while mitigating their weaknesses, making it particularly well-suited for $k$-step trajectory prediction. The explicit guidance from IL is expected to expedite convergence by steering the model toward optimal moves early in training, while the explorative capacity of RL should enable the model to learn from near-optimal actions and enhance its generalization across varied instances. Collectively, this combined approach is anticipated to yield superior performance in both convergence speed and final solution quality compared to models trained exclusively with either RL or IL.

# 3 Research Questions

To rigorously test our hypothesis, we propose to investigate the following research questions:

1. **Performance and Convergence:**
    - Does the hybrid RL-IL approach converge faster and produce higher quality solutions, compared to training solely with RL or IL?
2. **Loss Balancing Strategy:**
    - What is the optimal strategy for dynamically balancing the contributions of the RL and IL losses during training?
3. **Comparison with Conventional Local Search:**
    - How does the neural-based hybrid approach compare to traditional LS methods in terms of solution quality and computational cost?

# 4 Methods

Our method targets a combinatorial optimization problem in which a candidate solution is iteratively improved by a neural model that emulates a classical LS strategy. At each iteration, the model selects a move (for example, a bit flip in a binary representation of a solution to the Max Cut problem) that is expected to yield the greatest improvement in the objective function.

## 4.1 Hybrid RL-IL Framework

Our training framework combines two complementary components:

- **Imitation Learning (IL):** For each problem instance and a candidate solution, an optimal trajectory of $k$ steps is computed by exhaustive search. The first move from this trajectory is used as the target action, serving as a direct imitation signal.
- **Reinforcement Learning (RL):** Simultaneously, the model is trained with RL, where rewards are assigned based on the actual improvement in the objective function produced by the chosen moves.

The overall loss function is defined as:

$$L_{\text{total}} = \alpha_{IL} L_{\text{IL}} + \alpha_{RL} L_{\text{RL}},$$

where $L_{\text{IL}}$ is the imitation loss, $L_{\text{RL}}$ is the reinforcement learning loss, and $\alpha_{RL}$ and $\alpha_{IL}$ are the scaling factors that balance the contribution of the RL and IL losses, respectively.

## 4.2 Dynamic Gradient Normalization

To ensure that both the IL and RL components contribute effectively during training, we adopt a dynamic gradient normalization strategy. Specifically, for the current mini-batch, we compute the gradients with respect to the network parameters $\theta$ and determine their norms:

$$\|g_{\text{IL}}\| = \left( \sum_i \|g_{\text{IL},i}\|^2 \right)^{1/2}, \quad \|g_{\text{RL}}\| = \left( \sum_i \|g_{\text{RL},i}\|^2 \right)^{1/2}.$$

where $g_{\text{IL}} = \nabla_\theta L_{\text{IL}}$ and $g_{\text{RL}} = \nabla_\theta L_{\text{RL}}$ are the gradients computed with the IL and RL losses, respectively.

Then, we choose a target effective gradient norm $\mu$ for each component. Setting $\mu = 0.5$ ensures that when both gradients are combined, the total effective gradient norm is centered around 1.

To normalize the contributions, we scale each loss by a factor inversely proportional to its gradient norm. Specifically, we define:

$$\alpha_{IL} = \frac{\mu}{\|g_{\text{IL}}\| + \epsilon}, \quad \alpha_{RL} = \frac{\mu}{\|g_{\text{RL}}\| + \epsilon},$$

where $\epsilon$ is a small constant (e.g., $10^{-8}$) to avoid division by zero.

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
