# OpenReview forum: "Predicting $k$-Step Optimal Trajectories in Local Search"
_MAEB/2025/Projects_Track — MAEB 2025 Proyectos_

### Official Review · Reviewer_PM2h · 2025-03-17
**Integrating Reinforcement Learning and Imitation Learning for Trajectory Prediction in Local Search.**

**Rating:** 4
**Confidence:** 4

**Review:**

The authors clearly state their central hypothesis: a hybrid RL-IL framework will improve k-step trajectory prediction by enhancing both performance and convergence. The focus is on applying this approach to the Maximum Cut Problem, a well-known combinatorial optimization problem.

The experimental analysis is structured around three key research questions: whether the hybrid approach converges faster and produces higher-quality solutions compared to using RL or IL alone; investigating the optimal way to dynamically adjust RL and IL contributions during training; and evaluating how the hybrid method performs relative to traditional locar search techniques in terms of solution quality and computational efficiency. The experimental analysis seems well-structured, but further details on the evaluation metrics and baselines used for comparison would be valuable. In particular, it will be especially interesting to see how computational efficiency is measured and analyzed, as this will be a key factor in assessing the practical feasibility of the hybrid approach.

The topic is both interesting and relevant, particularly given the increasing role of machine learning in combinatorial optimization problems. Expanding on the hybrid framework’s implementation details and the evaluation criteria would strengthen the project. Looking forward to seeing how the experimental results align with the initial hypothesis.

## Pros:
* Clear hypothesis: the research is focus on integrating RL and IL for trajectory prediction in local search.
* Relevant domain: the research aligns with the growing interest in machine learning for combinatorial optimization.
* Application: the hybrid framework will be tested in the context of a well-known problem (Maximum Cut Problem).
* Structured research questions: the authors mention to focus on key aspects as performance, convergence, loss balancing, and comparison with local search.

## Cons:
* Lack of implementation details: based on the given information, it’s difficult to have a clear idea about the inner workings of the framework.
* Unclear evaluation metrics: further specification of how overall success will be measured (baselines, benchmarks) would strengthen the study.
* Computational efficiency considerations: a key concern is how computational efficiency will be assessed, which is crucial for practical feasibility.

---

### Official Review · Reviewer_M6oU · 2025-03-17
**Report on extended abstract**

**Rating:** 4
**Confidence:** 4

**Review:**

The propose neural models to predicting the optimal move after k local search steps in an effort to reduce local search computation.  They tested their approach on the Max Cut Problem. The problem and method is interesting and falls within the conference scope. Thus, I recommedn acceptance of the abstract.  I am assuming the experimental results in detail will be presented at the conference.

---

### Official Review · Reviewer_Egr3 · 2025-03-18
**Nice idea about local search computational cost saving.**

**Rating:** 5
**Confidence:** 5

**Review:**

Based on previous successful evidence, the project proposes to combine reinforcement learning with imitation learning in the prediction of the k-step trajectory. The proposal looks solid, and preliminary results would be interesting to be presented at the conference. Here are just some comments to improve it:
1) Section 3. The research questions might be focused on a particular combinatorial optimization problem. It is hard to generalize in combinatorial optimization (each optimization problem is an entirely different world), and such precision in the proposal will enrich its technical quality.
2) In Section 3, the authors talk about optimality. Is it to be proved theoretically?
3) Please include the performance indicators to be used to determine the improvement expected.
4) Including the number and type of test instances may be convenient.

---

### Decision · Program_Chairs · 2025-03-19

Accept